# Spin transport of a doped Mott insulator in moiré heterostructures

Emma C. Regan [1,2,3,12], Zheyu Lu[1,2,3,12], Danqing Wang[1,2,3,12], Yang Zhang [4,5,6], Trithep Devakul [4], Jacob H. Nie[7], Zuocheng Zhang [1], Wenyu Zhao[1], Kenji Watanabe [8], Takashi Taniguchi [9], Sefaattin Tongay [10], Alex Zettl[1,3,11], Liang Fu [4] ✉ & Feng Wang [1,3,11] ✉

Moiré superlattices of semiconducting transition metal dichalcogenide heterobilayers are model systems for investigating strongly correlated electronic phenomena. Specifically, $WSe_2/WS_2$ moiré superlattices have emerged as a quantum simulator for the two-dimensional extended Hubbard model. Experimental studies of charge transport have revealed correlated Mott insulator and generalized Wigner crystal states, but spin transport of the moiré heterostructure has not yet been sufficiently explored. Here, we use spatially and temporally resolved circular dichroism spectroscopy to directly image the spin transport as a function of carrier doping and temperature in $WSe_2/WS_2$ moiré heterostructures. We observe diffusive spin transport at all hole concentrations at 11 Kelvin − including the Mott insulator at one hole per moiré unit cell − where charge transport is strongly suppressed. At elevated temperatures the spin diffusion constant remains unchanged in the Mott insulator state, but it increases significantly at finite doping away from the Mott state. The doping- and temperature-dependent spin transport can be qualitatively understood using a $t$–$J$ model, where spins can move via the hopping of spin-carrying charges and via the exchange interaction.

Despite the complexity of strongly interacting quantum systems, the macroscopic dynamics of conserved quantities like spin and charge can often be described by emergent classical hydrodynamics. At low temperatures, transport properties are highly sensitive to the ground state and the nature of low-lying quasiparticles, long wavelength order parameter fluctuations, and the presence of superfluidity. At higher temperatures, transport can be understood in terms of the Nernst–Einstein relation arising from the diffusive spreading of conserved quantities. While both spin and charge are carried by electrons, the hydrodynamic coefficients governing spin and charge diffusion are generally distinct. Understanding how spin transport emerges from a microscopic model is an important question in studying quantum dynamics in strongly interacting quantum matter.

[1]Department of Physics, University of California at Berkeley, Berkeley, CA, USA. [2]Graduate Group in Applied Science and Technology, University of California at Berkeley, Berkeley, CA, USA. [3]Material Science Division, Lawrence Berkeley National Laboratory, Berkeley, CA, USA. [4]Department of Physics, Massachusetts Institute of Technology, Cambridge, MA, USA. [5]Department of Physics and Astronomy, University of Tennessee, Knoxville, TN, USA. [6]Min H. Kao Department of Electrical Engineering and Computer Science, University of Tennessee, Knoxville, TN, USA. [7]Department of Physics, University of California at Santa Barbara, Santa Barbara, CA, USA. [8]Research Center for Electronic and Optical Materials, National Institute for Materials Science, 1-1 Namiki, Tsukuba, Japan. [9]Research Center for Materials Nanoarchitectonics, National Institute for Materials Science, 1-1 Namiki, Tsukuba, Japan. [10]Department of Physics, Ma. School for Engineering of Matter, Transport and Energy, Arizona State University, Tempe, AZ, USA. [11]Kavli Energy NanoSciences Institute at University of California Berkeley and Lawrence Berkeley National Laboratory, Berkeley, CA, USA. [12]These authors contributed equally: Emma C. Regan, Zheyu Lu, Danqing Wang. ✉e-mail: liangfu@mit.edu; fengwang76@berkeley.edu

The moiré superlattice, formed by stacking two-dimensional (2D) materials with a relative twist angle or lattice mismatch, offers a unique platform for studying strongly correlated physics in a highly tunable setting[1–17]. Semiconductor transition metal dichalcogenide (TMD) moiré superlattices, such as $WSe_2/WS_2$, has emerged as a quantum simulator of the extended triangular lattice Hubbard model[1,2]. Crucially, the narrow moiré bandwidth allows access to a new parameter regime of doped Mott insulators, characterized by the separation of scales $t \ll k_B T \ll U$, typically inaccessible in traditional solid-state materials where $k_B T$ is usually much lower than $t$. Here $t$ is the hopping amplitude, $k_B T$ is the thermal energy, and $U$ is the onsite Coulomb repulsion. Measurements of charge transport in the $WSe_2/WS_2$ moiré superlattice have revealed correlated insulators, including Mott[1,2] and generalized Wigner crystal states[1], at specific filling of the superlattice unit cell. Yanhao et al., provided direct transport measurements showing resistance peak at Mott state[2]. Emma et al., provided an alternative approach named optically detected resistance and capacitance (ODRC) and showed a Mott insulating state as well[1]. Theories predict that spin physics, such as quantum spin liquids and superfluid spin transport, can also emerge in correlated states of moiré heterostructures[18–20]. While the presence of spin moments with antiferromagnetic interaction has been observed, experimental characterization of spin transport phenomena in moiré heterostructures has been challenging so far.

Here we use spatial- and temporal-resolved circular dichroism spectroscopy to directly image non-equilibrium spin transport in $WSe_2/WS_2$ moiré heterostructures, giving insight into the behavior of spin in a doped 2D Hubbard model.

## Results & Discussion

In the TMD moiré superlattice, the optical selection rules provide an opportunity to conveniently create and probe spin excitations using circularly polarized light[21–23]. A circularly polarized beam generates an exciton population in one valley, which is also spin-polarized due to the spin–valley locking in TMDs. Although this spin polarization is typically short-lived in TMD monolayers due to the electron-hole exchange interaction, long-lived spin lifetimes can be realized in type II heterostructures, like $WSe_2/WS_2$, where the electron and hole occupy separate layers[24]. Importantly, previous studies have shown that one can optically create a pure spin excitation, with no associated charge excitation, that has a lifetime of many microseconds in a hole-doped $WSe_2/WS_2$ heterostructure[1,24].

Using this approach, we optically generate a local spin population in the hole-doped $WSe_2/WS_2$ moiré superlattice and image their spatial and temporal evolution (Fig. 1b). We observe diffusive spin transport that is reduced compared to the spin transport in heterostructures without a large wavelength moiré superlattice. The spin transport remains significant at the Mott state with one hole per moiré unit cell ($p/p_0 = 1$) at 11 K, where charge transport is strongly suppressed[1,2]. This observation is consistent with spin-charge separation in the Mott state. Furthermore, the spin transport exhibit unusual temperature and doping dependence in the moiré heterostructure. This spin transport can be captured by a $t$–$J$ model on a triangular lattice, where spin conduction occurs via nearest neighbor hopping $t$ of spin-polarized free carriers and via exchange coupling $J$. At the Mott state, holes are immobile, so antiferromagnetic exchange between charges on neighboring superlattice sites is responsible for the spin diffusion. When additional electrons or holes are doped into the Mott state, the resulting empty and doubly occupied sites allow for conduction of spin-polarized holes.

Here we study the spin transport in a gated $WSe_2/WS_2$ moiré superlattice (device D1, see Supplementary Fig. 1). The $WSe_2$ and $WS_2$ monolayers are stacked with near-zero-degree twist angle and encapsulated in hBN. Few-layer graphite flakes form the electrostatic gate and contact, allowing for the injection of electrons and holes into the device. The ~4% lattice mismatch between the $WSe_2$ and $WS_2$ layers results in a moiré superlattice with a period of $a_M$ ~8 nm. Figure 1c shows the doping-dependent reflection contrast spectrum of device D1 in the energy range around the $WSe_2$ intralayer excitons. The three absorption peaks are signatures of the moiré superlattice[25], where the strong moiré potential results in three $WSe_2$ intralayer excitons states with different spatial characters. The lowest-energy peak is strongly enhanced when the heterostructure is doped to one electron or hole per moiré unit cell ($p_0 = 1.8 \times 10^{12}$ cm$^{-2}$, see Methods) due to the formation of Mott insulator states.

We use a spatial- and temporal-resolved pump–probe technique[24] to image spin transport in a hole-doped $WSe_2/WS_2$ moiré superlattice (Fig. 1b, Supplementary Fig. 2). We exploit the unusual ultrafast dynamic processes in the TMD heterostructures to achieve pure spin excitation with circularly polarized light in hole-doped $WSe_2/WS_2$ heterostructures[24]. Briefly, a circularly polarized pump at 1.807 eV selectively excites K valley excitons in the $WSe_2$ layer. Due to the spin–valley locking in TMDs, the K-valley excitons are composed of spin-up electrons and holes. The electrons transfer to the $WS_2$ layer within 100 fs, and their spins depolarize at the nanosecond time scale. The spin-depolarized electrons in the $WS_2$ layer can then recombine with holes of either spin in the $WSe_2$ layer over ~100 ns. After the recombination, net spin excitation of holes (i.e., more spin-up holes than spin-down holes) remains in the $WSe_2/WS_2$ moiré superlattice. The net spin polarization is equivalent to a net valley polarization in the TMD heterostructure due to the spin–valley locking. Therefore, the observed spin transport can also be viewed as valley transport. We

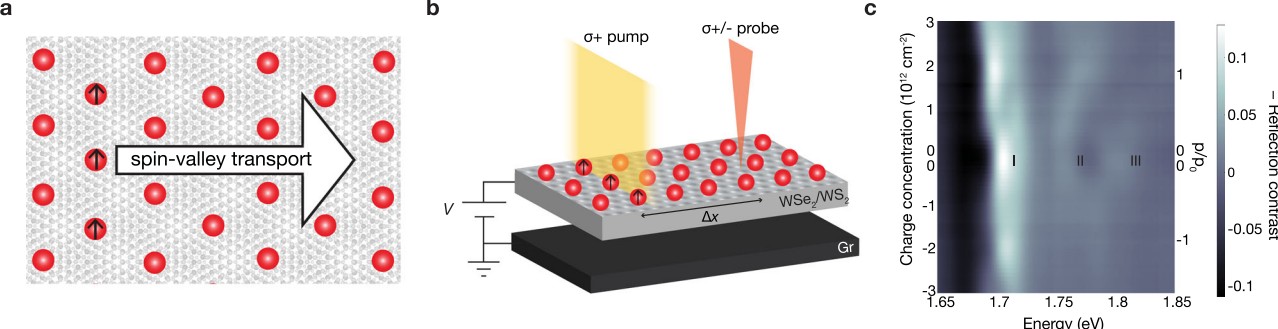

**Fig. 1 | Optical generation and detection of spin transport in a $WSe_2/WS_2$ moiré superlattice. a** An optically-generated 1D spin–valley polarization (up arrows) drives a spin current in the doped moiré superlattice (red circles are holes) at the Mott insulator state (one hole per moiré unit cell, $p/p_0 = 1$). **b** A circularly polarized pump beam (yellow) creates a local spin polarization in the hole-doped moiré superlattice. The time-resolved circular dichroism of a probe beam (orange)

measures the evolution of the 1D spin polarization in space and time as the pump–probe spatial separation ($\Delta x$) is scanned. A voltage V on the capacitor formed by the graphite gate and the moiré superlattice is used to tune the equilibrium hole concentration, p. **c** Doping-dependent reflection contrast spectrum of device D1, a $WSe_2/WS_2$ moiré superlattice. The three moiré exciton states are labeled as I, II, and III. The probe beam is near-resonant with exciton I.

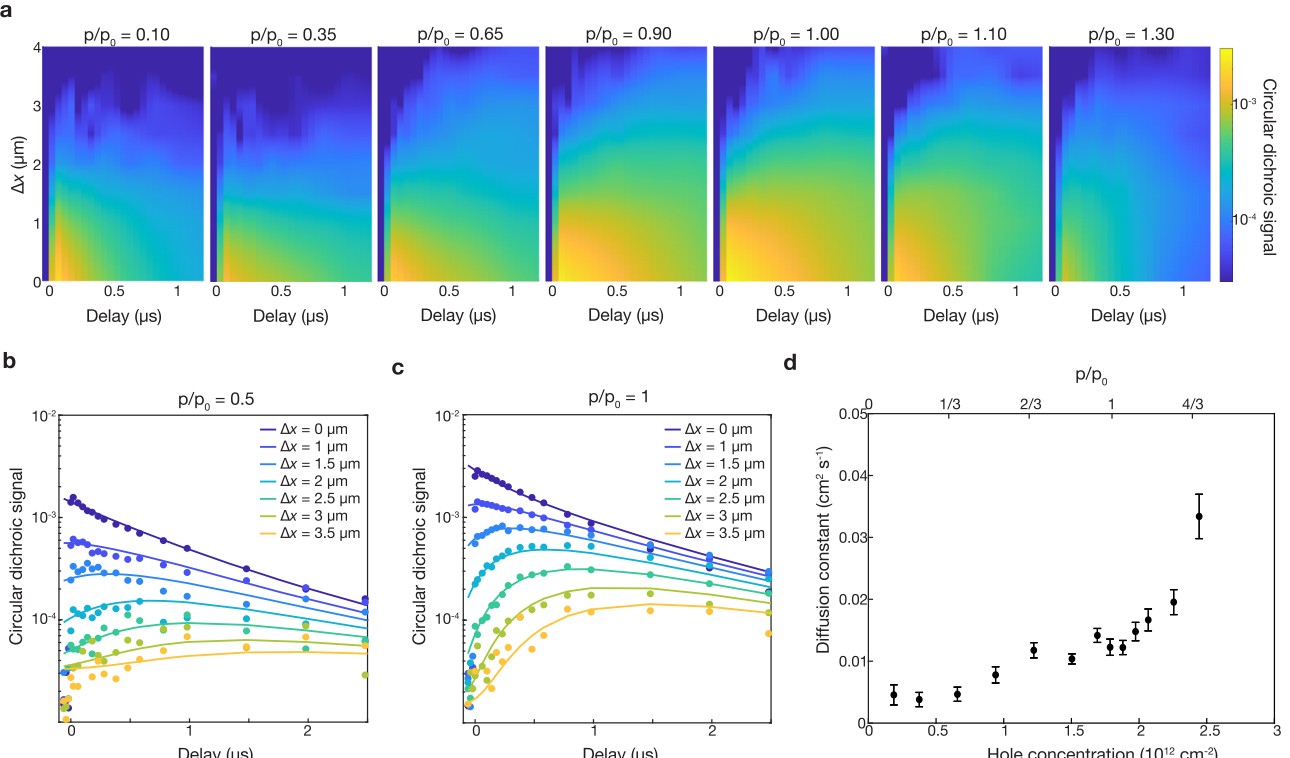

**Fig. 2 | Diffusive spin transport in a doped WSe₂/WS₂ moiré superlattice. a** The spatial-temporal evolution of a spin excitation in the WSe₂/WS₂ heterostructure at various hole concentrations, labeled in units of $p/p_0$. The horizontal and vertical axes represent the temporal and spatial separation between the pump and probe pulses, respectively. The color represents the circular dichroic signal on a log scale. **b**, **c** Horizontal cuts from **a** (dots) and fits the diffusion-decay model (lines) for $p/p_0 = 0.5$ (**b**) and $p/p_0 = 1$ (**c**). The spin transport is well described by a diffusion-decay model for all hole concentrations. **d** Extracted spin diffusion constant at various initial hole concentrations in the moiré superlattice. Error bars represent 95% confidence intervals obtained when fitting the data to a diffusion-decay model.

focus the pump beam into a line to create a one-dimensional (1D) spin excitation with an FWHM of approximately 1 μm that propagates through the sample. We probe the evolution of the spin polarization via the pump-induced circular dichroism of a probe beam at 1.703 eV, near-resonant with the lowest-energy WSe₂ exciton. By tuning the spatial separation ($\Delta x$) and the temporal delay between the pump and probe beams, we monitor the spin population in both space and time.

Using the spatial-temporal pump-probe technique, we generate and monitor a pure spin population in device D1 as we tune the charge concentration from near charge neutral to past the Mott state ($p/p_0 = 1$) at 11 K, well below the Mott transition temperature of ~150 K[2]. Figure 2a shows the evolution of the spin-polarized hole population for a subset of the hole concentrations measured (see Supplementary Fig. 3 for full dataset). The horizontal and vertical axes represent the temporal and spatial separation between the pump and probe pulses, respectively. At zero-time delay, the optically-generated spin polarization is localized at the pump position ($\Delta x < 1$ μm), and the circular dichroism is negligible away from the pump. After a finite time delay, the spin-polarized holes move out of the excitation region, and the circular dichroism increases away from the pump. Notably, spin transport is observed at all doping levels, including at $p/p_0 = 1$ where the system is electrically insulating. Therefore, the spin transport is decoupled from the charge transport in the Mott insulator at $p/p_0 = 1$.

Like in large-twist angle heterostructures[24], the spin transport in the moiré superlattice is diffusive for all measured hole concentrations. The spin-polarized hole density $\Delta p_s(x, t)$ can be described by a diffusion-decay model:

$$\Delta p_s(x, t) = \frac{\Delta p_0}{\sqrt{\pi(\sigma_0^2 + 4D_s t)}} e^{-\frac{x^2}{\sigma_0^2 + 4D_s t}} e^{-\frac{t}{\tau}} \qquad (1)$$

where $\Delta p_0$ is the total number of pump-induced spin-polarized holes, and $\sigma_0$ is the FWHM of the pump beam, $D_s$ is the spin diffusion constant, and $\tau$ is the spin lifetime. $x$ and $t$ are the space and time coordinates respectively. We fit the experiment data according to the diffusion-decay model based on the nonlinear least squares method. From the fitting results, we get the fitted parameters along with their confidence intervals. We achieve > 0.9 $R^2$ and <$10^{-4}$ root-mean-square error (RMSE) levels across all fittings which convinces the quality of our fittings (Supplementary Fig. 5 and Supplementary Table 1). Figure 2b and c show the measured circular dichroic (CD) signal (circles) and corresponding fits to the diffusion-decay model (lines) for $p/p_0 = 0.5$ and $p/p_0 = 1$, respectively. The doping-dependent spin lifetime has been reported[1], and our result is shown in Supplementary Fig. 6. At low hole concentration, the spin lifetime increases with doping because the spin lifetime becomes decoupled from the charge lifetime[24]. The spin lifetime is very long at the Mott insulator states[1], followed by a rapid decrease at higher doping. The detailed spin lifetime behavior shows some variation between different samples[1], possibly due to variations in defect density, residue strain, or inhomogeneities. Currently, there is no microscopic understanding of the spin lifetime behavior in moiré heterostructures. In this paper, we focus on the spin-diffusive transport.

Figure 2d shows the extracted spin diffusion constant at various hole concentrations in the superlattice at 11 K, which shows a smooth dependence on the hole concentration in the WSe₂/WS₂ moiré heterostructure. Notably, the spin diffusion constant does not exhibit a noticeable decrease at the Mott insulator state. This is in striking contrast to the electrical transport behavior of the moiré heterostructure, where pronounced suppression of conductance is observed at the Mott insulator state in previous studies[1,2]. It provides signatures of the decoupling of the spin and charge transport in the Mott

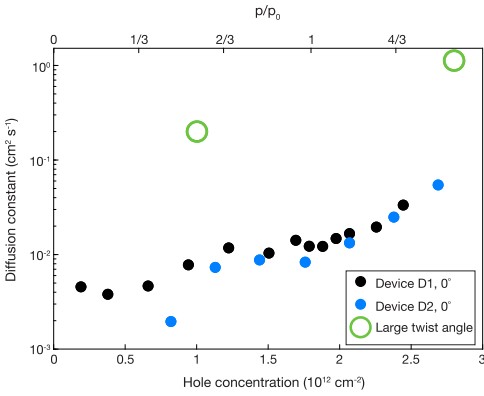

**Fig. 3 | Spin diffusion in aligned and misaligned WSe₂/WS₂ heterobilayers.** Extracted spin diffusion constant from two near-zero-twist angle moiré super-lattices (black and blue dots) and a large-twist-angle heterobilayer[24] (green circles). The spin diffusion is strongly suppressed in a moiré superlattice.

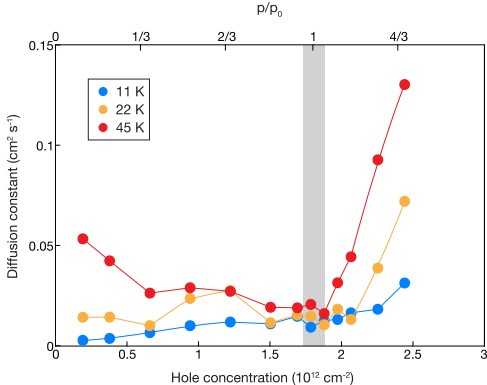

**Fig. 4 | Temperature-dependent spin diffusion.** The doping-dependent diffusion constants are extracted from spatial-temporal pump-probe measurements at three temperatures: 11 K, 25 K, and 45 K. The circles are fits to the diffusion model, and the lines are guides to the eye. The Mott insulator at $p/p_0 = 1$ is highlighted in grey. A weak temperature dependence is observed at the Mott insulator.

insulator state. The lack of suppression of spin diffusion constant at the Mott state indicates that the spin transport is dominated by spin-spin couplings rather than diffusion, and the effect from disorder-induced itinerant holes is small. Otherwise, one expects to observe a strong increase of the spin diffusion constant from the higher itinerant hole concentration away from the Mott density, similar to the increase of the charge conductance from the higher itinerant hole concentration away from the Mott density.

Figure 3 compares the doping-dependent spin diffusion constant for device D1 to a second near-zero twist angle WSe₂/WS₂ hetero-structure D2, and a large twist angle WSe₂/WS₂ heterostructure. The two near-zero twist angle moiré superlattice samples show similar spin diffusion constants for a given hole concentration, which are more than an order of magnitude smaller than that of the large twist angle heterostructure. The reduction of spin diffusion in the moiré super-lattice is a manifestation of strongly correlated electron states in the moiré superlattice flat bands.

We measure the doping-dependent spin transport at two elevated temperatures of 22 K and 45 K. The extracted spin diffusion constants are shown in Fig. 4. At the Mott state, highlighted in grey, the spin diffusion depends weakly on temperature. When temperature changes from 11 K to 22 K (100% change) and 45 K (309% change), the diffusion constant increases only by 11% and 53% respectively for the Mott state ($p/p_0 = 1$). When additional electrons or holes are doped to the Mott insulator, the spin diffusion becomes notably faster at elevated tem-peratures. For example, the spin diffusion constant increases by 130% and 317% respectively for $p/p_0 = 1.3$ and 296% and 1078% respectively for $p/p_0 = 0.2$ when the temperature increases from 11 K to 22 K and 45 K.

We can understand this doping and temperature-dependent spin transport in the moiré heterostructures through an effective Hubbard model on the triangular lattice[26]. For WSe₂/WS₂, the hopping ampli-tude $t$ is much smaller than the on-site Coulomb repulsion $U$[27]. In this regime, for $\nu \leq 1$, the Hubbard model reduces to the $t$–$J$ model, $H_{tJ} = -t\sum_{\langle ij\rangle, \sigma}(c_{i\sigma}^\dagger c_{j\sigma} + \text{h.c.}) + J\sum_{\langle ij\rangle}(\vec{S}_i \cdot \vec{S}_j - \frac{n_i n_j}{4})$ which captures the essential physics. Here $\nu$ is the filling factor of the moiré superlattice, $\vec{S}_i$ is the spin-$\frac{1}{2}$ operator given by $\vec{S}_i = \sum_{\sigma\sigma'} c_{i\sigma}^\dagger \frac{\sigma_{\sigma\sigma'}}{2} c_{i\sigma'}$ and $n_i$ is the particle number operator given by $n_i = \sum_\sigma c_{i\sigma}^\dagger c_{i\sigma}$, where $\sigma$ is the Pauli matrix, and $c_{i\sigma}$ are electron annihilation/creation operators on the site $i$. At higher density $\nu > 1$, the essential physics can again be captured by an effective $t$–$J$ model at filling $2 - \nu$, but with modified parameters.

Theoretical calculation of the spin diffusion constant is highly challenging in such systems but is possible under certain assumptions and limits[28–37]. We consider the spin diffusion constant from the $t$–$J$ model via the Nernst–Einstein relation $D_s = \sigma_s/\chi_s$, where $\sigma_s$ and $\chi_s$ are

the d.c. spin conductivity and compressibility in equilibrium. We obtain analytic expressions for $D_s$ in the high-temperature limit of the $t$–$J$ model (corresponding to the realistic separation of scales $t \ll k_B T \ll U$ of the Hubbard model) in two limits, given by

$$D_s^2 = \begin{cases} \frac{9\pi\delta^2 t^2 a^4}{24 - 20\delta - 4\delta^2} & J = 0 \\ \frac{9\pi J^2 a^4 \left(1 - 2\delta + \delta^2\right)}{16(7 - 5\delta)} & t = 0 \end{cases} \tag{2}$$

where $\delta = |\nu - 1|$ is the doping away from 1. Some physical insight can be gained from these expressions. For small $\delta$, we find $D_s/t \approx \sqrt{\frac{3\pi}{8}}\delta$ increases with $\delta$ in the $J = 0$ limit, which can be understood as an increase in the number of itinerant spin-current carriers. In the $t = 0$ limit, $D_s/J \approx \sqrt{\frac{9\pi}{112}}(1 - \frac{9}{14}\delta)$ instead decreases with $\delta$, which can be understood as a decrease in the number of local spin moments. At high temperatures, the observed increase of $D_s$ away from the Mott insulator indicates that the dominant contribution to spin transport is via the increase of itinerant spin carriers, in agreement with the theoretical expectation that $J/t = 4t/U \ll 1$.

We further address the general temperature and doping depen-dence through exact diagonalization, and indeed find that the observed trends can be described by realistic model parameters. We extract an exchange coupling $J \approx 0.1$ meV from the measured $D_s$ at the Mott insulator. Here, the weak temperature dependence and Einstein relation implies a $T$-linear spin resistivity $\rho_s = (D_s\chi_s)^{-1} \sim T$, since the spin compressibility $\chi_s \approx 1/(4T)$ for $T \gg J$. The asymmetric increase of $D_s$ upon doping away from the Mott insulator implies that the effective hopping amplitude $t$ is much larger for $\nu > 1$ than for $\nu < 1$. A possible explanation is that holes for $\nu > 1$ fill a charge transfer band[27], which can again be described by a $t$–$J$ model with an enhanced hopping through the physics of Zhang and Rice[38]. An alternative possible explanation is electron-assisted hopping due to Coulomb repulsion within a moiré site that effectively increases the size of the localized orbital and therefore the hopping integral[39].

The complex interplay of spin and charge physics in doped quantum magnets has received recent theoretical and experimental attention in the platform of ultracold atomic Hubbard model simulators[40]. Spin transport has been observed in 1D spin chains in the low[41,42] and high[43] temperature regimes, and in the 2D square lattice Hubbard model near half filling[28,44]. Here we have directly visualized spin in a highly non-equilibrium setting, approaching the level of control possible in ultracold atomic simulators but in a solid-state

platform. Conclusive demonstration and quantitative understanding of the spin-charge separation behavior could be achieved with further experimental studies, where simultaneous electrical and spin transport measurements can be performed on TMD heterostructures with different combinations of TMDs and twist angles.

## Methods
### Device fabrication
To fabricate the $WSe_2/WS_2$ moiré superlattices, we first exfoliate $WS_2$ and $WSe_2$ monolayers from bulk crystal onto $SiO_2/Si$ substrates and do polarization-resolved second harmonic generation (SHG) measurements to determine the crystal orientation in each flake. We then assemble the $WS_2$ and $WSe_2$ monolayers into a heterostructure with their crystal axes aligned using a polycarbonate (PC) stamp. During the transfer process, the near-zero-degree-twist-angle $WSe_2/WS_2$ stack is contacted by few-layer graphite (FLG) and sandwiched between two hexagonal boron nitride (hBN) layers with thickness of 15-25 nm. Additional FLG flakes serve as electrostatic gates. The whole heterostructure is released onto a 90 nm $SiO_2/Si$ substrate. Electrodes (100 nm Au with 5 nm Cr adhesion layer) are fabricated using a photolithography system (Durham Magneto Optics, MicroWriter) and an electron-beam deposition system. After fabrication, we again perform polarization-resolved SHG measurements on the monolayer $WS_2$ and $WSe_2$ regions within the heterostructure to determine the exact twist angle and on the heterostructure region to distinguish between near-zero and near-sixty-degree samples.

### Doping-dependent spatial-temporal pump–probe technique
A function generator (Siglent SDG6022X) is used to generate two synchronized electronic pulse trains that drive two RF-coupled laser diode modules (Thorlabs LDM56) with center wavelengths of ~690 nm (pump) and ~730 nm (probe). The probe laser wavelength is finetuned to be near-resonant with the lowest energy $WSe_2$ A exciton using the temperature-controlled mount. The pump beam travels through a polarizer and shared quarter waveplate (QWP) to generate a left circularly polarized pump. The probe beam travels through a polarizer, rotating half-wave plate (HWP), and the shared QWP to switch between left- and right- circularly polarized probe. The pump and probe beams are combined with a dichroic mirror and focused onto the sample mounted in a Montana Instruments cryostation using an objective. An additional cylindrical lens in the pump path ensures that the pump beam is focused on a line on the sample. The pump–probe spatial separation is tuned via a piezo-controlled mirror in the pump path. For all pump-probe measurements, the pump power is set such that the photoexcited hole population is small compared to the electrostatically injected hole population.

The reflected beams are spectrally filtered with a 715 nm long-pass filter to isolate the probe, which is then monitored with an avalanche photodiode (APD). The APD output is sent to a Keithley 2400 SourceMeter (measures the static reflectivity, RC) and a SRS864A lock-in amplifier (measures the pump-induced change in the reflectivity, ΔRC). The lock-in amplifier is locked to the frequency of the optical chopper in the pump path.

Doping dependent spin transport measurements are conducted by applying voltages to the graphite gate using Keithley 2400 or 6482 SourceMeters. The charge density $p$, is defined using a parallel plate capacitor model:

$$p = \pm \frac{1}{e}\frac{\epsilon_{hBN}\epsilon_0}{d}(V - V_0) \qquad (3)$$

where $\epsilon_{hBN}$ is the dielectric constant of hBN (measured as $4.2 \pm 0.2$), $\epsilon_0$ is the permittivity of free space, $d$ is the thickness of the bottom hBN layer, and $V$ is the voltage applied to the gate. We account for the quantum capacitance (voltage range where charges are not injected)

using an offset voltage $V_0$ for electron and hole doping. The offsets are defined by the voltage where the reflection contrast spectrum begins to change with electron and hole doping.

The moiré density $p_0$ is defined to correspond to one hole per moiré unit cell and is determined through the relation $p_0 = 1/[L_M^2 \sin(\pi/3)]$ where $L_M$ is the moiré superlattice constant. The twist angle ($\theta$) and lattice mismatch ($\delta = (a - a')/a$) between the two layers determines $L_M$ via $L_M = a/\sqrt{\delta^2 + \theta^2}$. STM measurements of a near-zero degree $WSe_2/WS_2$ give $L_M \sim 8$ nm, which is consistent with that expected due to the lattice constant mismatch between the layers. Therefore, for zero-degree-twist angle samples, $p_0 = 1.80 \times 10^{12}$ cm$^{-2}$.

## Data availability
The main data that support the findings of this study are available within the article and its Supplementary Information files. More supporting data are available from the corresponding authors upon request. Source data are provided with this paper.

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

## Acknowledgements

This work was supported primarily by the Director, Office of Science, Office of Basic Energy Sciences, Materials Sciences and Engineering Division, of the US Department of Energy under contract no. DE-AC02–05-CH11231, within the van der Waals Heterostructures Program (KCWF16), which provided for the design of the project, device fabrication, and spin transport experiments. Additional support was provided by the Director, Office of Science, Office of Basic Energy Sciences, Materials Sciences and Engineering Division, of the US Department of Energy under contract no. DE-AC02-05-CH11231, within the Nanomachines Program (KC1203), which provided for supplementary microscopy. The work at Massachusetts Institute of Technology was supported by the U.S. Department of Energy, Office of Science, Basic Energy Sciences, under award no. DE-SC0020149. S.T. acknowledges support from DOE-SC0020653, NSF DMR 2111812, NSF DMR 1552220, NSF 2052527, DMR 1904716, and NSF CMMI 1933214 for WS2 bulk crystal growth and analysis. K.W. and T.T. acknowledge support from the JSPS KAKENHI (Grant Numbers 19H05790 and 20H00354).

## Author contributions

F.W. conceived the research. E.C.R., Z.L. and D.W. carried out optical measurements. E.C.R., Z.L., D.W. and F.W. performed data analysis. Y.Z., T.D. and L.F. performed theoretical calculations. E.C.R., Z.L., D.W., J.N., Z.Z., W.Z. and A.Z. contributed to the fabrication of van der Waals heterostructures. S.T. grew WSe2 and WS2 crystals. K.W. and T.T. grew hBN crystals. All authors discussed the results and wrote the manuscript.

## Competing interests

The authors declare no competing interests.
