## [Transparent Peer Review file · Nature Communications]

REVIEWER COMMENTS

Reviewer #3 (Remarks to the Author):

In the manuscript by E. C. Regan et al., the authors investigate the spin transport of holes in gate-tunable, angle-aligned WSe₂/WS₂ bilayer hosting deep, triangular moire superlattice potential. The experiments are carried out in a very similar way to that introduced by the overlapping group of authors in Ref. [24] to study spin diffusion in angle-misaligned WSe₂/WS₂ heterostructures. Specifically, a circularly-polarized and elliptically-shaped pump pulse is used to locally excite hole spins in a narrow spatial region of the sample. Subsequent relaxation and transport of such excited hole spins are analyzed by monitoring time-resolved circular polarization degree of the probe pulse reflected off the device at different spatial locations. By fitting the results of such experiments with a simple diffusion model, the authors extract hole-density-dependent spin diffusion constant D_s and spin relaxation time at three different temperatures. At the lowest temperature ($T=11$ K), D_s does not show any significant drop around the unity filling factor ($\nu = 1$) of the moire lattice, where the holes have been proven by numerous previous studies to form a Mott insulating state exhibiting substantially reduced conductivity. Based on this striking difference between the present spin transport data and prior charge transport results around $\nu = 1$, the authors argue that the spin-charge transports are decoupled from each other and that the spin transport in the Mott state is primarily driven by exchange interactions and not by charge diffusion. This conclusion is further supported by the analysis of t-J model predictions, which allows the authors to roughly estimate the inter-site hopping and exchange energies.

First, I agree with the two former Reviewers that, while the conclusions of this work are sound and exciting, the experimental evidence could have been stronger. More specifically, I concur with Reviewer #1 that it would be much more elegant to perform charge and spin transport measurements on the same device. At the same time, it is clear that such experiments would require a separate device (e.g., partially-covered with a top gate electrode enabling contactless probing of optically-detected resistance as in Ref. [1]) that would be homogeneous enough to allow for reliable spin transport sensing. Although assembly of such a device does not seem to be out of reach of present 2D fabrication technology (as proven, for example, by recent transport observation of a fractional Hall resistance of a fractional Chern insulator in a twisted MoTe₂ bilayer), I understand the authors point that it certainly represents an experimental challenge.

In addition, I also agree with the authors that suppression of charge transport around $\nu = 1$ can be regarded as relatively well-established characteristic of this type of bilayer system. It is thus implausible that this suppression does not occur in the presently investigated device, especially given that its optical response, as shown in Fig. 1c, is very similar to that observed in prior studies (including the prominent increase of the intensity of the lowest-energy moire exciton peak around $\nu = 1$). Still, without a direct measurement, it is certainly not clear whether the charge transport is fully suppressed upon the formation of the Mott state. In fact, given, e.g., the twist angle disorder, it is actually quite unlikely. Yet, unlike the Reviewer #1, I do not consider this to be a necessary condition to make a claim about (at least partial) spin-charge separation. In my opinion, this is revealed by strikingly different temperature response of charge and spin transport: while the charge conductivity at $\nu = 1$ increases upon rising the temperature faster than that for $\nu \neq 1$ (as seen, e.g., in Ref. [2]), the spin transport at $\nu = 1$ is demonstrated by the present study to be almost temperature-independent, although at $\nu \neq 1$ it increases with the temperature. This qualitatively different behavior clearly shows that the spin transport in the Mott state is sizably decoupled from the charge transport.

Taking into account the above arguments, and given that the investigated phenomenon is of high interest to the research community working on correlated physics in 2D materials, I feel that this manuscript might be worth publication. However, in my opinion, there are several important points that should be addressed/revised by the authors beforehand.

1) In view of the above discussion, I suggest to slightly tone down the claim that the spin and charge transport is completely decoupled (e.g., like in the statement “It clearly demonstrates the decoupling of the spin and charge transport in the Mott insulator state”).

2) While the doping dependence of the spin diffusion constant is discussed in the main text, the spin relaxation time is only analyzed in the SI (section S5). Although the authors claim that its “behavior is consistent with previous observations”, the increase of the spin lifetime at $\nu = 1$ is barely visible (particularly with respect to $\nu < 1$ side). This is in quite a clear contrast to what was seen by the overlapping group of authors in Ref. [1], where the spin relaxation time at $\nu = 1$ was by at least 30% longer as compared to its value at $\nu = 1 \pm 1/3$. What is the reason for this difference? Is it related to a larger disorder of the present device? This is a very important point that should be definitely clarified and highlighted in the main text.

3) Can the authors assess how homogenous the optical response of the present sample is? (for example by mapping the exciton energy or the hole density corresponding to $\nu = 1$ across the device area that is analyzed in the spin transport experiments). This could considerably strengthen the claim of negligible role of disorder-induced effects.

4) Has it been verified that the spin current scales linearly with the optical excitation power?

5) How does a clear asymmetry in hopping energy between $\nu < 1$ and $\nu > 1$ sides (revealed by fitting t-J model to temperature-dependent spin transport data) correspond to rather symmetric charge transport seen in Refs [1,2]. Is this a property of the presently investigated device?

3rd Round Reviewer's comments:

Reviewer #3 (Remarks to the Author):

In the manuscript by E. C. Regan et al., the authors investigate the spin transport of holes in gate-tunable, angle-aligned WSe₂/WS₂ bilayer hosting deep, triangular moiré superlattice potential. The experiments are carried out in a very similar way to that introduced by the overlapping group of authors in Ref. [24] to study spin diffusion in angle-misaligned WSe₂/WS₂ heterostructures. Specifically, a circularly polarized and elliptically-shaped pump pulse is used to locally excite hole spins in a narrow spatial region of the sample. Subsequent relaxation and transport of such excited hole spins are analyzed by monitoring time-resolved circular polarization degree of the probe pulse reflected off the device at different spatial locations. By fitting the results of such experiments with a simple diffusion model, the authors extract hole-density-dependent spin diffusion constant D_s and spin relaxation time at three different temperatures. At the lowest temperature ($T=11$ K), D_s does not show any significant drop around the unity filling factor ($\nu = 1$) of the moiré lattice, where the holes have been proven by numerous previous studies to form a Mott insulating state exhibiting substantially reduced conductivity. Based on this striking difference between the present spin transport data and prior charge transport results around $\nu = 1$, the authors argue that the spin-charge transports are decoupled from each other and that the spin transport in the Mott state is primarily driven by exchange interactions and not by charge diffusion. This conclusion is further supported by the analysis of t-J model predictions, which allows the authors to roughly estimate the inter-site hopping and exchange energies.

First, I agree with the two former Reviewers that, while the conclusions of this work are sound and exciting, the experimental evidence could have been stronger. More specifically, I concur with Reviewer #1 that it would be much more elegant to perform charge and spin transport measurements on the same device. At the same time, it is clear that such experiments would require a separate device (e.g., partially covered with a top gate electrode enabling contactless probing of optically-detected resistance as in Ref. [1]) that would be homogeneous enough to allow for reliable spin transport sensing. Although assembly of such a device does not seem to be out of reach of present 2D fabrication technology (as proven, for example, by recent transport observation of a fractional Hall resistance of a fractional Chern insulator in a twisted MoTe₂ bilayer), I understand the authors point that it certainly represents an experimental challenge.

In addition, I also agree with the authors that suppression of charge transport around $\nu = 1$ can be regarded as relatively well-established characteristic of this type of bilayer system. It is thus implausible that this suppression does not occur in the presently investigated device, especially given that its optical response, as shown in Fig. 1c, is very similar to that observed in prior studies (including the prominent increase of the intensity of the lowest-energy moiré exciton peak around $\nu = 1$). Still, without a direct measurement, it is certainly not clear whether the charge transport is fully suppressed upon the formation of the Mott state. In fact, given, e.g., the twist angle disorder, it is actually quite unlikely. Yet, unlike the Reviewer #1, I do not consider this to be a necessary condition to make a claim about (at least partial) spin-charge separation. In my opinion, this is revealed by strikingly different temperature response of charge and spin transport: while the charge conductivity at $\nu = 1$ increases upon rising the temperature faster than

that for $\nu < 1$ (as seen, e.g., in Ref. [2]), the spin transport at $\nu = 1$ is demonstrated by the present study to be almost temperature-independent, although at $\nu < 1$ it increases with the temperature. This qualitatively different behavior clearly shows that the spin transport in the Mott state is sizably decoupled from the charge transport.

Taking into account the above arguments and given that the investigated phenomenon is of high interest to the research community working on correlated physics in 2D materials, I feel that this manuscript might be worth publication. However, in my opinion, there are several important points that should be addressed/revised by the authors beforehand.

1) In view of the above discussion, I suggest to slightly tone down the claim that the spin and charge transport is completely decoupled (e.g., like in the statement “It clearly demonstrates the decoupling of the spin and charge transport in the Mott insulator state”).

Response 1:

We thank the reviewer for the comment.

The referee suggested to slightly tone down the original claim that the spin and charge transport is clearly decoupled. We have followed the referee’s suggestion to remove any strong claim on spin-charge separation in the abstract and introduction. The revised comparison between charge and spin transport in the introduction paragraph is the following (line 163 - 167):

Notably, the spin diffusion constant does not exhibit a noticeable decrease at the Mott insulator state. This is in striking contrast to electrical transport behavior of the moiré heterostructure, where pronounced suppression of conductance is observed at the Mott insulator state in previous studies^{1,2}. It provides signatures of the decoupling of the spin and charge transport in the Mott insulator state.

2) While the doping dependence of the spin diffusion constant is discussed in the main text, the spin relaxation time is only analyzed in the SI (section S5). Although the authors claim that its “behavior is consistent with previous observations”, the increase of the spin lifetime at $\nu = 1$ is barely visible (particularly with respect to $\nu < 1$ side). This is in quite a clear contrast to what was seen by the overlapping group of authors in Ref. [1], where the spin relaxation time at $\nu = 1$ was by at least 30% longer as compared to its value at $\nu = 1 \pm 1/3$. What is the reason for this difference? Is it related to a larger disorder of the present device? This is a very important point that should be definitely clarified and highlighted in the main text.

Response 2:

We thank the reviewer for the question.

The referee asked questions about the behavior of spin relaxation time as a function of doping in the WSe₂/WS₂ moiré heterostructure. In [Nature **579**, 359-363 (2020)], Fig. 4c shows a clear peak of the spin lifetime at $\nu=1$ with the order of 8 microseconds and it decreases significantly on both sides when we dope away from $\nu=1$. As a comparison, in the current paper, we have a

spin lifetime on the order of 2 microseconds and we see a rapid decrease when $\nu > 1$. The only difference is that we do not see a rapid decrease when $\nu < 1$.

Currently there is no theoretical understanding of the microscopic mechanics that control the spin relaxation lifetime in moiré heterostructures. It is possible that this is related to local defect density or some inhomogeneity in different devices. We have added the following discussions in the revised manuscript on the spin relaxation behavior (line 153 - 160):

The doping-dependent spin lifetime has been reported^{1,25}, and our result is shown in Fig. S6. At low hole concentration, the spin lifetime increases with doping because the spin lifetime becomes decoupled from the charge lifetime²⁴. The spin lifetime is very long at the Mott insulator states¹, followed by a rapid decrease at higher doping. The detailed spin lifetime behavior shows some variation between different samples^{1,25}, possibly due to variations in defect density, residue strain, or inhomogeneities. Currently, there is no microscopic understanding of the spin lifetime behavior in moiré heterostructures. In this paper, we focus on the spin diffusive transport.

3) Can the authors assess how homogenous the optical response of the present sample is? (for example, by mapping the exciton energy or the hole density corresponding to $\nu = 1$ across the device area that is analyzed in the spin transport experiments). This could considerably strengthen the claim of negligible role of disorder-induced effects.

Response 3:

We thank the reviewer for the question.

The referee asked questions about the homogeneity of the WSe₂/WS₂ moiré heterostructure device. We got the hole density corresponding to $\nu = 1$ across the device by measuring the doping dependent moiré trion intensity as a function of doping across different regions of the device (in a 2-3-3 grid, 8 locations in total). The mapping region ($\sim 10 \times 10 \mu\text{m}^2$) is already much larger than our real measurement region where our maximum spatial extent is $3.5 \mu\text{m}$. The results are shown in Figure R1.

We chose 0.05V for our voltage step. The hole density of $\nu = 1$ corresponds to the local maximum of the trion intensity. We used the Savitzky-Golay filter with window size 15 and polynomial order 3 to smooth out the curve. The choice of 15 as window size makes the smoothing less sensitive to local variation and helps to get the correct trend. We then use a polynomial function with order 6 to fit the filtered curve, get the maximum argument (Mott voltage/density), and use bootstrap to get the standard error. From the figure, we could see that the Mott voltage/density is bounded between -2.53 V and -2.68 V over the mapping region: The backgate voltage fluctuation for the Mott insulator state is within 0.15 V. For a 20 nm thick hBN, a 0.15 V gate change corresponds to $\sim 0.18 \times 10^{12} \text{ cm}^{-2}$ density change, which is much smaller than moiré density in the system $\sim 1.80 \times 10^{12} \text{ cm}^{-2}$).

The trion intensity offset do change a bit due to local field effect in the absorption spectrum. However, it does not influence our spin diffusion measurement since we are measuring the circular dichroism signal which is less sensitive to the local field variation across the device.

Figure R1. Trion intensity as a function of doping and the voltage corresponding to the Mott insulating state ($\nu = 1$).

4) Has it been verified that the spin current scales linearly with the optical excitation power?

Response 4:

We thank the reviewer for the question.

The referee asked questions about the justification of the pump power used in the experiment. We used 686 nm with 180 nW for the cylindrical-shaped pump beam. We chose this based on former pump-probe measurement where such power lies in the linear regime. If we further lower the power, the signal-to-noise ratio will decrease. Therefore, we chose 180 nW as a sweet spot for the pump beam power.

In addition, as a rough estimation, the exciton density resulting from such power can be estimated using the rate equation and typical lifetime of moiré excitons in TMD systems [Nature Physics **19**, 1286-1292 (2023)]. For an excitation energy of 1.81 eV (686 nm), each photon carries an energy of 2.9×10^{-19} J. We use a pump power of 180 nW on $\gg 4 \mu\text{m}^2$ area with a power intensity of $P \ll 0.045 \mu\text{W} \mu\text{m}^{-2}$, there are 0.15×10^{20} photons $\text{s}^{-1} \text{cm}^{-2}$ incident on the sample. Using an absorptance of 0.02, we can estimate that around 3.0×10^{17} excitons $\text{s}^{-1} \text{cm}^{-2}$ are created. The decay rate is on the order of $\Gamma = (100 \text{ ns})^{-1}$. The rate equation reads $dN/dt = \alpha - \Gamma N = 0$ at steady state. Therefore, the exciton density can be approximated as $N = \alpha/\Gamma = 0.03 \times 10^{12} \text{ cm}^{-2}$. This value is much smaller than the hole density $1.8 \times 10^{12} \text{ cm}^{-2}$ corresponding to $\nu = 1$ in our WSe₂/WS₂ moiré superlattice which justifies our choice of power.

5) How does a clear asymmetry in hopping energy between $\nu < 1$ and $\nu > 1$ sides (revealed by fitting t-J model to temperature-dependent spin transport data) correspond to rather symmetric charge transport seen in Refs [1,2]. Is this a property of the presently investigated device?

Response 5:

We thank the reviewer for the question.

The referee asked questions about the relation between our fitted t-J model parameters and the charge transport behavior of the moiré heterostructure as a function of doping. In both Fig. 2 of [Nature **579**, 353-358 (2020)] and Fig. 2 of [Nature **579**, 359-363 (2020)], the transport curve is not symmetric between $\nu < 1$ and $\nu > 1$. This asymmetric charge transport behavior seen in transport [Nature **579**, 353-358 (2020), Nature **579**, 359-363 (2020)] between $\nu < 1$ and $\nu > 1$ is consistent with our observation of asymmetric spin transport in the sample.

REVIEWERS' COMMENTS

Reviewer #1 (Remarks to the Author):

In the revised version of the manuscript, the authors have addressed the concerns I raised in my original review. In particular, I appreciate the authors' efforts to tone down their initial claims on the evidence of spin-charge separation. However, I also find some answers to be not thorough enough. This particularly concerns the point about experimental verification of linear scaling between the optical excitation power and the measured spin current. When addressing this point, the authors provided reasonable arguments justifying their choice of the excitation power. However, the fact that the exciton density generated at this particular power level is much lower than the electron density does not prove that the system really remains in the linear regime. To demonstrate this, one would need to determine the spin current for at least a few different powers and then check whether it increases linearly or not. While such a measurement clearly lies within the experimental capabilities of the authors, it has not been carried out even though it could strengthen the conclusions drawn in the manuscript.

The second potential weakness concerns prominent asymmetry in the filling-factor dependence of the spin lifetime on $\nu < 1$ and $\nu > 1$ sides that has not been clearly observed in the previous study by the same group of authors. Although I agree with the authors that this deviation could be, to some extent, blamed on different disorder in the investigated devices, one should be careful with following this line of argumentation given that it is also used in completely opposite way when discussing the charge-transport at $\nu = 1$. In fact, the latter quantity is assumed to follow the same trend as in the previous studies independently of the conjectured differences in the disorder level. Taking this into account, I would find it more convincing if the present experiments could be reproduced on a separate device (which is in general a good practice in the field of moire materials).

In view of the above, I feel that the manuscript could still be improved. At the same time, it seems to me that the authors could have made already enough to justify its publication. Nevertheless, I leave the final decision on this point to the editors.

4th Round Reviewer's comments:

Reviewer #1 (Remarks to the Author):

In the revised version of the manuscript, the authors have addressed the concerns I raised in my original review. In particular, I appreciate the authors' efforts to tone down their initial claims on the evidence of spin-charge separation. However, I also find some answers to be not thorough enough. This particularly concerns the point about experimental verification of linear scaling between the optical excitation power and the measured spin current. When addressing this point, the authors provided reasonable arguments justifying their choice of the excitation power. However, the fact that the exciton density generated at this particular power level is much lower than the electron density does not prove that the system really remains in the linear regime. To demonstrate this, one would need to determine the spin current for at least a few different powers and then check whether it increases linearly or not. While such a measurement clearly lies within the experimental capabilities of the authors, it has not been carried out even though it could strengthen the conclusions drawn in the manuscript.

The second potential weakness concerns prominent asymmetry in the filling-factor dependence of the spin lifetime on $\nu < 1$ and $\nu > 1$ sides that has not been clearly observed in the previous study by the same group of authors. Although I agree with the authors that this deviation could be, to some extent, blamed on different disorder in the investigated devices, one should be careful with following this line of argumentation given that it is also used in completely opposite way when discussing the charge-transport at $\nu = 1$. In fact, the latter quantity is assumed to follow the same trend as in the previous studies independently of the conjectured differences in the disorder level. Taking this into account, I would find it more convincing if the present experiments could be reproduced on a separate device (which is in general a good practice in the field of moire materials).

In view of the above, I feel that the manuscript could still be improved. At the same time, it seems to me that the authors could have made already enough to justify its publication. Nevertheless, I leave the final decision on this point to the editors.

Reply: We thank the referee's comments.

First, we note that experimental data from two devices are shown in Figure 3 of the main text. Both devices show a monotonic increase of diffusion constant as a function of the carrier density, therefore an asymmetry in spin diffusion behavior on $\nu < 1$ and $\nu > 1$.

We agree with the referee that further experimental studies could establish the spin-charge separation behavior more conclusively. We have added the following sentence to the conclusion of the manuscript. "Conclusive demonstration and quantitative understanding of the spin-charge separation behavior could be achieved with further experimental studies, where simultaneous electrical and spin transport measurements can be performed on TMD heterostructures with different combinations of TMDs and twist angles."